# Protective Role of Social Networks for the Well-Being of Persons with Disabilities: Results from a State-Wide Cross-Sectional Survey in Kerala, India

**DOI:** 10.3390/ijerph20054213

**Published:** 2023-02-27

**Authors:** Saju Madavanakadu Devassy, Lorane Scaria, Shilpa V. Yohannan, Sunirose Ishnassery Pathrose

**Affiliations:** 1Department of Social Work, Rajagiri College of Social Sciences (Autonomous), Rajagiri P.O., Kalamassery, Kochi 683 104, India; 2Rajagiri International Centre for Consortium Research in Social Care, Rajagiri College of Social Sciences (Autonomous), Kochi 683 104, India; 3Department of Social Work, Melbourne School of Health Sciences, The University of Melbourne, Melbourne, VIC 3010, Australia; 4Rajagiri Research Institute, Rajagiri College of Social Sciences (Autonomous), Kochi 683 104, India

**Keywords:** people with disabilities, service accessibility, social networks, mental health, wellbeing, Kerala

## Abstract

The current study presents the findings from a cross-sectional survey on social factors associated with the well-being of persons with disabilities (PWDs) in Kerala, India. We conducted a community-based survey across three geographical zones, North, Central, and South of Kerala state, between April and September 2021. We randomly selected two districts from each zone using a stratified sample method, followed by one local self-government from each of these six districts. Community health professionals identified individuals with disabilities, and researchers collected information on their social networks, service accessibility, well-being, and mental health. Overall, 244 (54.2%) participants had a physical disability, while 107 (23.78%) had an intellectual disability. The mean well-being score was 12.9 (S.D = 4.9, range = 5–20). Overall, 216 (48%) had poor social networks, 247 (55%) had issues regarding service accessibility, and 147 (33%) had depressive symptoms. Among the PWDs with issues with service access, 55% had limited social networks. A regression analysis revealed that social networks (b = 2.30, *p* = 0.000) and service accessibility (b = −2.09, *p* = 0.000) were associated with well-being. Social networks are more important than financial assistance because they facilitate better access to psycho-socioeconomic resources, a prerequisite for well-being.

## 1. Introduction

### 1.1. People with Disabilities in Kerala, India

Disabilities and related complications, irrespective of their types, pose severe challenges across the globe. Globally, more than 15% of people live with a disability, and the prevalence is significantly higher among people from low- and middle-income countries than in other developed countries [1]. According to the disability census of Kerala, there are 793,937 people with disabilities in Kerala, which accounts for 2.32% of the total population [2], where the national average is 2.21% [3]. Among the different types of disabilities in Kerala, locomotor disability is the most common type, accounting for 31% of the total PWDs, followed by multiple disabilities accounting for 17%, and mental illness (12%). Vision and hearing impairment accounts for 7.8% and 7.6%, respectively. Further, 46.63% of PWDs in Kerala are living below the poverty line [2]. In low- and middle-income countries like India, the rapid increase in disability incidence and severity has not been accompanied by planned initiatives to enhance their well-being and overall health [4]. Due to various systemic barriers, the meager welfare services and programs already available are only accessed by a small proportion of people [5,6].

### 1.2. Social Networks of People with Disabilities

People with disabilities generally experience low levels of social integration and inclusion compared to the general population [7,8] for various reasons, such as functional limitations [9], social stigma, and discrimination [10]. People with disabilities have fewer social contacts and are less likely to begin relationships in everyday life [11], further leading to poorer employment opportunities and health outcomes [12]. Moreover, a study on different social networks among people in Kerala showed that 37.6% of PWDs had a private restricted network type rather than a locally integrated one [13]. All of these reasons clubbed together can cause an increased risk of social isolation among this already vulnerable group [14,15]. Further, people with disabilities have been found to have higher odds of depression and anxiety levels [13], and the personal and health characteristics of PWDs have been found to be mediated by social cohesion in Kerala [16].

In countries like India, where resource scarcity weakens social security nets, family members and neighbors should play a crucial role in the care and support of PWDs [17]. Neighborhood connectivity has the potential to provide knowledge from network members about locally accessible formal and informal resources, effective interventions, health behaviors, and employment opportunities [18]. PWDs create their networks based on employment, routine activities, family connections [19,20], and neighborhood interactions [21,22]. In unequal societies with weak safety nets, this networking is vital for learning about available resources, preventing the loss of existing services, lobbying for additional welfare measures, ensuring greater access to resources locally [23], and creating more growth opportunities. The existing evidence shows that more cohesive societies cooperate in providing welfare services to meet the needs of PWDs, mainly through resource mobilizations at the societal level [24]. Moreover, PWDs feel identified with a group or neighborhood that accepts and is compassionate towards them, which increases their social status [24], and, consequently, their mental health [25].

Family and neighborhood are the best sources of support for PWDs, given the scarcity of social support measures and the overall collectivist nature of Indian societies. However, there is a dearth of evidence about the specific social factors associated with the well-being of those with disabilities. We assume that developing a sense of connectedness and inclusion would play a pivotal role in enhancing their well-being which would moderate the negative impact of disabilities. The findings of this study will help practitioners and policymakers in India to devise strategies focused on strengthening social networking and neighbourhood connectivity to enhance the well-being of these people.

## 2. Materials and Methods

### 2.1. Design

We conducted a cross-sectional, community-based study of PWDs across three geographical zones—North, Central, and South of Kerala state, India—between April and September 2021. Kasaragod, Wayanad, Kannur, Kozhikode, and Malappuram districts make up the Northern zone. The Central zone consists of four districts: Palakkad, Thrissur, Ernakulam, and Idukki. The Southern zone includes Trivandrum, Kollam, Pathanamthitta, Alappuzha, and Kottayam districts. We randomly selected six districts from these three zones (two from each) using a stratified sampling method, followed by selecting one local self-government (LSG) body from each of these six districts. The local self-government bodies are administrative divisions within each district that function as sub-units of each district. The LSGs include municipalities or corporations (sub-units in urban areas) and panchayats (rural areas). We randomly selected two units from urban areas (one corporation and one municipality). Four Panchayats (more panchayats were included to ensure better representation. (The Kerala state has 941 grama panchayats, 87 municipalities, and 6 corporations). Accredited Social Health Activists (ASHAs), who have an advantage due to their domicile, helped identify people with disabilities. After listing the names of the PWDs who had been identified, researchers made home visits until they had 75 consenting PWDs (or, in the case of children or those with severe disabilities, their carers) from each selected local self-government. Figure 1 describes the participant recruitment procedures of the current study.

### 2.2. Participant Recruitment

We recruited PWDs and their caregivers from the community through a multistage recruitment procedure. The researchers included the PWDs residing in the targeted location who consented to participate. We included people within the four major disability categories, including physical disability, intellectual disability, multiple disabilities, and other forms of disabilities. A random number technique was employed to identify 75 PWDs from each district and recruit a total of 450 participants for the current study.

### 2.3. Measurements

#### 2.3.1. Outcome Variable

The primary outcome measure of well-being was measured by the WHO Well-Being Index [21], which is a set of five questions measured on a Likert scale with response options of “all of the time” (5), “most of the time (4), “more than half of the time” (3), “less than half of the time” (2), “some of the time” (1) and “at no time” (0). The scores ranged between 0 and 25, and a higher score indicated better well-being. The tool has been validated and found to have good reliability coefficients [26].

#### 2.3.2. Exposure Variables

Sociodemographic variables, mental health, well-being, and access to services were the major exposure variables measured in the current study. Sociodemographic variables included age, gender, education, marital status, employment, the color of the ration card, the type(s) of disability, and the percentage level of disability. Age was ascertained in years and was later grouped into four categories: children (0–18 years), young adults (19–39 years), middle adulthood (40–59 years), and elderly (above 60 years). Education was measured in five categories: not literate, literate but did not complete primary education, completed primary education (10th grade), completed secondary education (12th grade), and completed tertiary and above (graduation, diploma, or post-graduation). Marital status was ascertained in four categories: currently married, never married, widowed, and divorced/separated. Occupational details were measured as “employed”, “unemployed”, “student”, or “completely dependent”. A ration card is an official document, issued by the state government, that describes the eligibility to purchase subsidized food grains from the government distribution system. The colors, coded as yellow, pink, blue, and white, describe the socio-economic status of each household. The yellow and pink cards are for households below the poverty line, while the blue and white cardholders fall above the poverty line. The types of disabilities were categorized into four areas: physical disability, including locomotor disability, vision, hearing, and speech impairment; intellectual disability, including mental retardation and autism; multiple disabilities; and other forms of disabilities, which included disabilities due to a chronic neurological condition, Parkinson’s, or mental illness. The percentage of disability is ascertained from the disability certificate issued by the Government of India.

Mental health was measured using the DASS 21 (Depression, Anxiety, and Stress) Scale [27]. It includes 21 self-reported questions rated on a four-point scale (0–3), with “0” denoting “did not apply to me at all” and “3” meaning “applied to me very much, or most of the time”. The DASS 21 is a reliable and valid tool to measure mental health among adults [28].

Access to services was measured using a set of self-reported questions based on accessibility in four major areas: family income/employment, essential services, health care, and mental health care. Accessibility was rated on a four-point scale (1–4), with “1” denoting “as much as I need”, “2” representing “most times”, “3” indicating “sometimes,” and “4” meaning “not at all”. We also asked self-reported questions about barriers to accessing care in four major areas: awareness, absence of services, lack of support, and transportation, to which the participants replied using binary response options of “yes” (1), denoting the presence of the barrier, and “no” (0), indicating an absence.

Social networks were measured using a set of self-reported questions about the level of contact and support received from families, friends, and neighbors. The questions were measured on a four-point Likert scale (0–3), with 0 denoting “at no time”, 1 denoting “sometimes,” 2 denoting “most times”, and 3 denoting “at all times”. Based on median scores, they were classified as people with poor social networks and people with adequate social networks for analysis purposes.

### 2.4. Data Analysis

We performed descriptive statistics to profile the PWDs concerning their geographical locations and other demographic variables. We calculated frequencies and percentages through two-way tables to find differences between the subgroups of interest. Further, Chi-square tests were used to determine the statistical difference between the variables. Linear regression was performed to identify the various factors associated with well-being among people with disability. The level of statistical significance was set at *p* < 0.05. All statistical analyses were performed in IBM SPSS 26 package (New York, NY, USA) and STATA (StataCorp LLC Version 15, Lakeway Drive, TX, USA).

### 2.5. Ethical Considerations

We obtained ethical committee approval from the institution’s Institutional Review Board (Ref. No. –RCSS/IEC/002/2021, dated 15 January 2021). We obtained informed written consent from participants and their caregivers before inclusion. We also explained the voluntary nature of participation and the right to withdraw at any data collection stage.

## 3. Results

### 3.1. Demographic Characteristics

The study included data from 450 respondents (Table 1), the majority of whom were males (62%). More than 65% of the respondents were in the early/or middle adulthood stage, and 12.9% were elderly. Overall, 72% of the respondents had completed primary education, while 3% were uneducated/illiterate. Further, 63% of the respondents were unmarried, 54% were unemployed/entirely dependent on family members, and 71% were below the poverty line. Of the types of disability, 54.2% had a physical disability, which included a multitude of disabilities related to vision, hearing, speech, or locomotor functioning, and 24% of the population had intellectual disabilities.

The mean well-being score for the study population was 12.9 (±4.9). There was no significant difference in well-being scores within the demographic variables studied. However, the scores were slightly higher for children, females, people who completed secondary or tertiary education, and people with less than 40% disability. Summative scores for depression, anxiety, and stress, measured by the DASS scale in the current study group, were 6.62 (6.3), 9.3 (8.7), and 8.2 (7.7), respectively. Further, 147 (32.7%) of PWDs had mild or above depression, 93 (20.7%) had mild or above anxiety, and 278 (61.8%) had mild or above stress.

Demographically, PWDs without formal education existed at the highest rates in the northern zone of Kerala, while unemployment among PWDs was the highest in the southern zone. The summative scores of well-being were the highest among PWDs in the south zone (mean = 14.2), followed by the north (mean = 13) and the central (mean = 11.7) zones. Mental illness, in terms of depression, anxiety, and stress, was the highest among PWDs in the central zone. Social support from neighbors and family members was comparatively higher in the southern zone than in others.

### 3.2. Service Accessibility

We studied access to income/employment, food, medical health care, and mental health care to study the service accessibility among PWDs. In the current population, there were many (40%) who could not access income-generating employment or medical services (25%). In contrast, most had access to essential services (94%), and 86% had access to mental health treatment. Service access in all areas was comparatively higher among males. Furthermore, access to income and essential services was relatively higher among PWDs residing in the southern parts of Kerala. In comparison, access to treatment was better in the northern parts compared to other zones (Table 2).

Of the 450 participants, 203 (45.11%) PWDs had no issue accessing services. However, among 247 people with service access issues, 149 (33.11%) had trouble accessing one service, 64 (14.22%) had issues with two services, 27 (6%) PWDs with three services, and 7 (1.56%) PWDs with all the services listed. Among the 247 PWDs with service access issues, 55% had limited social networks. However, among people with adequate service access, 60% had adequate social networks.

Table 3 describes the subgroup analysis of the accessibility variables with social networks and the types of disabilities. Inadequacies in accessing employment, essential services, medical care, and mental health care were more prevalent in people without adequate support from their families and neighborhoods. Overall, 62% of respondents having inadequate employment (statistically significant at *p* = 0.000), 59% of respondents having insufficient access to food/other essential services, 52% of respondents having inadequate medical health care, and 54% of respondents having poor access to mental health care had lower social network scores.

Table 4 presents the results of a linear regression analysis conducted to understand the association between social networks and well-being among the respondents. In the current study, people with adequate support were found to have 2.3 times higher scores for well-being compared to people with poorer social networks. The inability to access services and the presence of depression, anxiety, and stress symptoms were negatively associated with well-being in the current population.

## 4. Discussion

The current study aimed at identifying the role of social networks and other social factors in improving the well-being of people with disabilities. Demographically, PWDs with locomotor disabilities were the most common type, and the northern zone of Kerala had the largest percentage of PWDs without a formal education. In contrast, the southern zone had the highest rate of PWDs who were unemployed. Study results point to a comparatively lower number of people accessing services in the central geographical zone of Kerala. Although more of the PWDs in the southern zone were unemployed, those in the northern zone had less schooling. The center zone, which performed well in both of these areas, had poorer levels of well-being and a greater demand for mental health services, especially due to limited access to disability services. This can be explained by the fact that these zones are home to a predominantly urban population with lower neighborhood connectivity and linkages [29]. The current study findings suggest that poor neighborhood connectedness leads to limited access to information, and thereby, to services. This finding is in line with another study conducted in South India [30].

Furthermore, the study findings stressed the importance of family and neighborhood support networks for better well-being and protection against adverse health outcomes in PWDs [31]. People with adequate support networks in the study had higher scores for overall well-being, and this is consistent with studies elsewhere [32,33,34]. This is all the more critical in unequal and stratified, resource-poor societies like India, which is characterized by inadequate safety nets and lean spending on social welfare, such as health care, education, and unemployment insurance [4].

Tapping into local neighborhoods’ physical, social, and service facilities depends on the neighborhood’s culturally defined friendliness/helpfulness and support patterns. There is sufficient evidence to prove that people living in supportive communities require fewer mental health services [35] due to better well-being. The supportive neighborhood disseminates knowledge about self-care and promotes access to locally available services, amenities, and affective support [36]. Through an amalgamation of collective efficacy, social support, and the prevalence of local organizations and voluntary associations, this social connectedness improves access among PWDs [37]. The affective or cognitive closeness with others makes it easier for people to communicate their concerns and gain knowledge about resources [38], especially regarding the non-governmental and volunteer organizations that can address their needs [39]. The participants’ enhanced ability to obtain resources from their networks significantly increases their well-being. Due to their social disconnect, people with disabilities are frequently deprived of opportunities for inclusion, which has an impact on their well-being.

If social networking is created with cultural sensitivity, and in accordance with the current community ecosystem, it can enhance inclusion, social functioning, and resource linkages. The advancement of technology would be another way to increase connectedness and enlarge the borders of the neighborhood. A few digital networking models are worth experimenting with in order to enhance their social inclusion, learn about the resources available, and also advocate for legislation to promote access and social inclusion [40]. This study challenges the current focus of policymakers and practitioners, who emphasize financial support alone as a means of enhancing the well-being of PWDs. Social networks can potentially address the problems with inclusion, accessibility, and emotional requirements, indicating that PWDs are moving up the ladder of Maslow’s hierarchy of needs. This upward trend could be linked to the general economic progress of caring families in conjunction with the nation’s development. The findings encourage policymakers to undergo a paradigm shift in the target areas of intervention strategies. It should be firmly founded in defense of the rights, dignity, and self-worth of people with disabilities from a psycho-socioeconomic perspective as opposed to only an economic one. The current gaps in the care of PWDs could be filled by co-creating social networks, simplifying the linking pathways, and devising customized interventions.

The current study has its limitations as well. Firstly, this being a cross-sectional study, the observed associations cannot be interpreted as causal inferences. The study only included PWDs identified through community health workers and included only the known cases, which can limit the generalizability of the findings. Disability, a complex multidimensional phenomenon, cannot be fully measured quantitatively, which might be another limitation of the current study. However, the study’s findings encourage researchers to investigate PWDs lived experiences, particularly in light of the nation’s evolving psychosocial and economic environment.

## 5. Conclusions

Social networks and support are particularly crucial, as even the already existing formal and informal services and resources for these groups are embedded within the systems of society. The lack of a formal networking platform through which to meet each other in an empathetic environment is a significant barrier to accessing various resources. The well-being of people with disabilities would be improved by developing supportive neighborhood communities and including PWDs through participatory approaches. Social support and social networking take precedence over financial support because they give people a sense of belonging to a community and make it easier for them to obtain information about formal and informal services, eventually enhancing their well-being. The evidence of social networks and connectivity in enhancing the well-being of PWDs compels researchers to devise strategies to scale up the networking by utilizing technological advancements, such as mHealth, social media, and geospatial resource navigation facilities to detect, register, monitor, and link them, further facilitating customized service access to the respective community-dwelling PWDs.

## Figures and Tables

**Figure 1 ijerph-20-04213-f001:**
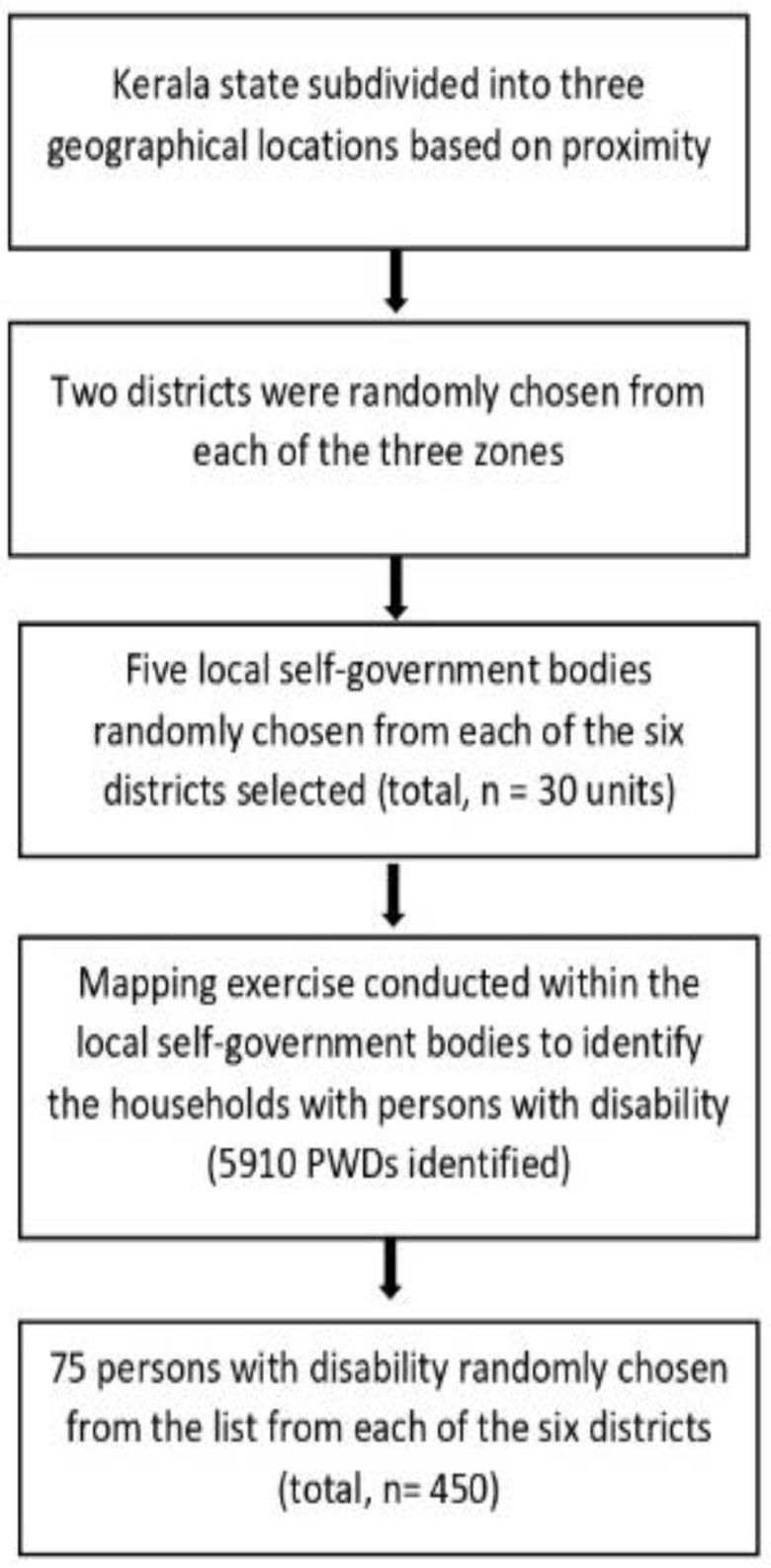
Participant recruitment.

**Table 1 ijerph-20-04213-t001:** Demographic Distribution of PWDs.

Variables	Category	Overall	Northern Zone	Central Zone	Southern Zone
Age	Children (0–18 years)	93 (20.7%)	28 (18.6%)	29 (19.3%)	36 (24%)
	Young adults (19–39 years)	151 (33.6%)	55 (36.7%)	54 (36%)	42 (28%)
	Middle adulthood (40–59 years)	148 (32.9%)	49 (32.7%)	57 (38%)	42 (28%)
	Late adulthood (>60 years)	58 (12.9%)	18 (12%)	10 (6.7%)	30 (20%)
Gender	Male	279 (62%)	91 (60.7%)	100 (66.7%)	88 (58.7%)
	Female	171 (38%)	59 (39.3%)	50 (33.3%)	62 (41.3%)
Education	Illiterate	74 (16.4%)	30 (20%)	22 (14.7%)	22 (14.7%)
	literate but did not complete primary education	14 (3.1%)	10 (6.7%)	1 (0.7%)	3 (2%)
	Completed primary education	324 (72%)	100 (66.7%)	114 (76%)	110 (73.3%)
	Completed secondary education	33 (7.3%)	9 (6%)	11 (7.3%)	13 (8.7%)
	Completed tertiary education	5 (1.1%)	1 (0.7%)	2 (1.3%)	2 (1.3%)
Marital status	Currently married	147 (32.7%)	52 (34.7%)	45 (30%)	50 (33.3%)
	Never married	284 (63.1%)	9 1(60.7%)	100 (66.7%)	93 (62%)
	Widowed	10 (2.2%)	4 (2.7%)	1 (0.7%)	5 (3.3%)
	Separated/Divorced	9 (2%)	3 (2%)	4 (2.7%)	2 (1.3%)
Employment	Employed	98 (21.8%)	32 (21.3%)	44 (29.3%)	22 (14.7%)
	Unemployed	96 (21.3%)	20 (13.3%)	10 (6.7%)	66 (44%)
	Student	109 (24.2%)	38 (25.3%)	41 (27.3%)	30 (20%)
	Dependent	147 (32.7%)	60 (40%)	55 (36.7%)	32 (21.3%)
Type of ration card	Yellow	64 (14.2%)	17 (11.3%)	20 (13.3%)	27 (18%)
	Pink	260 (57.7%)	94 (62.7%)	83 (55.3%)	19 (12.7%)
	Blue	58 (12.9%)	16 (10.7%)	23 (15.3%)	19 (12.7%)
	White	68 (15.1%)	23 (15.3%)	24 (16%)	21 (14%)
Type of disability	Physical disability	244 (54.2%)	69 (46%)	83 (55.3%)	92 (61.3%)
	Intellectual disability	107 (23.8%)	37 (24.7%)	33 (22%)	37 (24.7%)
	Multiple disabilities	69 (15.3%)	32 (21.3%)	23 (15.3%)	14 (9.3%)
	Other disabilities*	30 (6.7%)	12 (8%)	11 (7.3%)	7 (4.7%)
Level of disability	Below 40%	33 (7.3%)	9 (6%)	6 (4%)	18 (12%)
	Between 40–79%	314 (69.8%)	94 (62.7%)	116 (77.3%)	104 (69.3%)
	80% above	103 (22.9%)	47 (31.3%)	28 (18.7%)	28 (18.7%)
Well-being	Total mean score	12.94 (4.9)	13 (4.9)	11.7 (4.9)	14.2 (4.7)
Depression	Total mean score	6.6 (6.3)	6.9 (6.9)	8.1 (6.2)	4.8 (5.2)
Anxiety	Total mean score	9.3 (8.7)	9.2 (8.6)	11.2 (8.9)	7.4 (8.2)
Stress	Total mean score	8.2 (7.7)	8.2 (7.7)	10.1 (8.1)	6.2 (6.8)
Social Support	Poor support networks	216 (48%)	71 (47.3%)	85 (56.7%)	60 (40%)
	Adequate support networks	234 (52%)	79 (52.7%)	65 (43.3%)	90 (60%)

**Table 2 ijerph-20-04213-t002:** Subgroups of service accessibility by gender and geographical zone.

Service Type	with Access	Gender		Zone		
		Male	Female	North	Central	South
Income/Employment	271 (60.2%)	160 (59%)	111 (41%)	85 (31.4%)	69 (25.5%)	117 (43.1%)
Access to food/other daily services	421 (93.6%)	258 (61.3%)	163 (38.7%)	142 (33.7%)	133 (31.6%)	146 (34.7%)
Access to medical health care	336 (74.7%)	210 (62.5%)	126 (37.5%)	123 (36.6%)	108 (32.1%)	105 (31.3%)
Access to mental health care treatment.	386 (85.8%)	244 (63.2%)	142 (36.8%)	139 (36%)	127 (32.9%)	120 (31.1%)

**Table 3 ijerph-20-04213-t003:** Service accessibility and social support networks.

Accessibility Variables	Overall	Social Support Networks		*p*-Value
		Poor Support Networks	Support Networks Available	
General Services				
Employment access—adequate	271 (60.22%)	105 (38.75%)	166 (61.25%)	*p* < 0.001
Employment access—Inadequate	179 (39.78%)	111 (62.1%)	68 (37.9%)	
Food/basic services—adequate	421 (93.56%)	199 (47.27%)	222 (52.73%)	*p* = 0.238
Food/basic services—Inadequate	29 (6.44%)	17 (58.6%)	12 (41.4%)	
Medical health care—adequate	336 (74.67%)	156 (46.43%)	180 (53.57%)	*p* = 0.252
Medical health care—Inadequate	114 (25.33%)	60 (52.6%)	54 (47.4%)	
Mental health care—adequate	386 (85.78%)	181 (46.89%)	205 (53.11%)	*p* = 0.248
Mental health care—Inadequate	64 (14.22%)	35 (54.7%)	29 (45.3%)	

**Table 4 ijerph-20-04213-t004:** Associative factors of well-being among people with disabilities.

Associative Factors	Categories	Crude Regression Coefficient (CI), *p*-Value
Social Support Networks	Poor support networks (reference)	
	Support networks available	2.30 (1.40–3.19), *p* < 0.001
Education	(1-unit increase)	0.71 (0.18–1.24), *p* = 0.008
Service Accessibility	(no accessibility issues as reference)	
	Inability to access one of the services	−1.50 (−2.52 to −0.49), *p* = 0.004
	Inability to access two of the services	−2.08 (−3.42 to −0.73), *p* = 0.003
	Inability to access three of the services	−4.50 (−6.43 to −2.58), *p* < 0.001
	Inability to access all four services	−5.52 (−9.13 to −1.91), *p* = 0.003
Presence of depressive symptoms	(1-unit increase)	−0.65 (−0.69 to −0.61), *p* < 0.001
Presence of anxiety symptoms	(1-unit increase)	−0.49 (−0.52 to −0.47), *p* < 0.001
Presence of stress symptoms	(1-unit increase)	−0.55 (−0.58 to −0.52), *p* < 0.001

## Data Availability

The data supporting the findings of the study will be made available upon request to the corresponding author.

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
