# Peer review of "Protective Role of Social Networks for the Well-Being of Persons with Disabilities: Results from a State-Wide Cross-Sectional Survey in Kerala, India"

_ijerph, 2023, doi:10.3390/ijerph20054213_

Round 1
Reviewer 1 Report
Protective role of social networks for the well-being of persons with disabilities: Results from a state-wide cross-sectional survey in Kerala, India
International Journal of Environmental Research and Public Health
This is a purely descriptive study listing many demographic variables pertaining to people with disabilities. As such, there are few methodological issues.
There are several conceptual issues however. First and foremost is the disconnect between the title of the manuscript that promises a focus on the function of social networks in facilitating well-being among People With Disabilities (PWDs), and the thrust of the analyses as well as the first line of the Discussion section: “The current study aimed at identifying the socio-economic factors associated with the well-being of those with disabilities.”
Table 3 seems quite important, comparing those with poor vs. adequate support networks given the title of the manuscript. However, some of the sample sizes of the subgroups (i.e., those with “other disabilities”) are too small for subgroup comparisons to be made, as is the case with those without food/basic services (n=29) and conclusions to be drawn. In fact, Lines 201-211 draw conclusions regarding differences between those with and without adequate social supports, but other than those with inadequate employment there are no inferential statistics given. Looking at the differences between percentages of only the overall sample (not subdivided by type of disability), it is difficult to conclude that those other differences mentioned (ex: medical health care) are statistically significant (52.6% of those with poor vs. 47.4% of those with adequate supports). Rather than attempt to imply significant differences between those with and without adequate support networks, particularly listed within subgroups with small sample sizes, it would be much more helpful to accurately describe what was found if the true aim of the manuscript is to describe how social support networks help people with disabilities.
Table 4: I do not understand why the presence of depressive, anxiety, and stress symptoms are used as predictors of well-being when their very definition implies a lack of well-being? Rather, the issue at hand is the presence/absence of social support networks; it seems to me that the presence of these symptoms should function as variables that are predicted by presence/absence of social networks rather than predictors of well-being. Using actual scores for these variables rather than presence/absence of symptoms would allow linear regression to predict DSS scores using the presence/absence of social support networks as the predictor variable.
In summary, I found the manuscript to be without a clear focus. If the title of the manuscript promises to signal the story to come, then the focus needs to be on the usefulness of social networks to people with disabilities. The data are there, however they need to be presented in a much more coherent fashion to be useful.
Issues that need clarification/editing:
The category “other disabilities” should provide examples of what was included should the authors decide to continue to subdivide those with disabilities into separate categories.
How was percentage of disability arrived at? This concept is introduced in Line 169 but I can find no prior mention of it.
Lines 172 and 173 both list “mild and above depression” – most likely a type and I’m guessing that the latter should read stress.
p-levels should never be described 0 but rather as < .01, .001,.001, etc.
Author Response
Author Response to Reviewer 1
This is a purely descriptive study listing many demographic variables pertaining to people with disabilities. As such, there are a few methodological issues.
Author Response: Thank you for agreeing to review our manuscript.
There are several conceptual issues, however. First and foremost is the disconnect between the title of the manuscript that promises a focus on the function of social networks in facilitating well-being among People With Disabilities (PWDs), and the thrust of the analyses as well as the first line of the Discussion section: “The current study aimed at identifying the socio-economic factors associated with the well-being of those with disabilities.”
Author Response: revisions have been made to the manuscript to make it clearer.
Table 3 seems quite important, comparing those with poor vs. adequate support networks given the title of the manuscript. However, some of the sample sizes of the subgroups (i.e., those with “other disabilities”) are too small for subgroup comparisons to be made, as is the case with those without food/basic services (n=29) and conclusions to be drawn. In fact, Lines 201-211 draw conclusions regarding differences between those with and without adequate social supports, but other than those with inadequate employment there are no inferential statistics given. Looking at the differences between percentages of only the overall sample (not subdivided by type of disability), it is difficult to conclude that those other differences mentioned (ex: medical health care) are statistically significant (52.6% of those with poor vs. 47.4% of those with adequate supports).
Rather than attempt to imply significant differences between those with and without adequate support networks, particularly listed within subgroups with small sample sizes, it would be much more helpful to accurately describe what was found if the true aim of the manuscript is to describe how social support networks help people with disabilities.
Author response: Table 3 has been reconstructed, p values has also been added.
Table 4: I do not understand why the presence of depressive, anxiety, and stress symptoms are used as predictors of well-being when their very definition implies a lack of well-being? Rather, the issue at hand is the presence/absence of social support networks; it seems to me that the presence of these symptoms should function as variables that are predicted by presence/absence of social networks rather than predictors of well-being. Using actual scores for these variables rather than presence/absence of symptoms would allow linear regression to predict DSS scores using the presence/absence of social support networks as the predictor variable.
Author response: Table 4 is redone based on the suggestions.
In summary, I found the manuscript to be without a clear focus. If the title of the manuscript promises to signal the story to come, then the focus needs to be on the usefulness of social networks to people with disabilities. The data are there; however, they need to be presented in a much more coherent fashion to be useful.
Author response: The manuscript has been edited to make the focus clearer.
Issues that need clarification/editing:
The category “other disabilities” should provide examples of what was included should the authors decide to continue to subdivide those with disabilities into separate categories.
Author response: Clarifications made (line 125)
How was the percentage of disability arrived at? This concept is introduced in Line 169 but I can find no prior mention of it.
author response: Percentage of disability is elicited from the disability certificate – a certificate issued to PWDs in India describing the type and level of disability. A sentence has been added to measurements.
Lines 172 and 173 both list “mild and above depression” – most likely a type and I’m guessing that the latter should read stress.
author response: Corrections made
p-levels should never be described 0 but rather as < .01, .001,.001, etc.
author response: Corrections made
Reviewer 2 Report
Thank you very much for giving me the chance to review this very interesting article, dealing with a crucial issue: the relationship between persons with disabilities and social networks.
The study presents the findings from a cross-sectional survey on social and economic factors associated with the well-being of persons with disabilities in Kerala, India.
Although my knowledge about India is limited I believe that the methodology is appropriate.
The study conducted a community-based survey across three geographical zones, North, Central, and South of Kerala state, randomly selecting two districts from each zone using a stratified sample method, followed by one local self-government from each of these six districts.
Also the choice of allowing Community health professionals to identify individuals with disabilities, mixing participants With a physical an intellectual 22 disability are optimal choices.
Thus from a methodological perspective the article is sound.
Therefore it surely represents a necessary contribution to the stream of literature looking at social network uses by disabled people.
However, I believe that this work present one major flaws that the authors should fix: the article lacks a solid theoretical background
1) INTRODUCTION
The Introduction is somehow superficial and does not offer a solid background to justify the study.
I suggest dividing it in at least two subheading: the first dedicated –as you already do- to data about people with disabilities, maybe adding specificities of Kerala, to better justify the study.
The second should be dedicated to the wide and SEMINAL literature dealing with social networks and disability, that in the current state of the paper is not dealt with.
This should be the backbone of your paper that actually a) justify your work b) located in a stream of research
In particular, I strongly encourage you to situate your study within the framework DIGITAL AND MEDIA LITERACY, since disability can be a form of social exclusion (as your own study demonster) that can be tackled by media literacy.
These studies might help:
Friesem, Y. (2017). Beyond accessibility: How media literacy education addresses issues of disabilities. Journal of Media Literacy Education, 9(2), 1-16.
Cervi, L.; Tornero, J.M.P. Changing the Policy Paradigm for the Promotion of Digital and Media Literacy. The European Challenge.In Pursuing Digital Literacy in Compulsory Education: Reconstructing the School to provide Digital Literacy for All; Peter Lang Inc.: NewYork, NY, USA, 2011; pp. 50–70.
2) CONCLUSIONS
Once the theoretical background has been enriched in the Conclusion it would be important to engage with previous studies.
For instance, the rich social media and Covid-19 stream of literature might be used to both compare your results and underline their importance.
Some examples:
Tejedor, S.; Cervi, L.; Pérez-Escoda, A.; Tusa, F. Smartphone usage among students during COVID-19 pandemic in Spain, Italyand Ecuador. In Eighth International Conference on Technological Ecosystems for Enhancing Multiculturality, Salamanca, Spain, 21–23 October 2020; ACM: New York, NY, USA, 2020; pp. 571–576.
Dobransky, K., & Hargittai, E. (2021). Piercing the pandemic social bubble: Disability and social media use about COVID-19. American Behavioral Scientist, 65(12), 1698-1720.
GOOD LUCK!
Author Response
Thank you very much for giving me the chance to review this very interesting article, dealing with a crucial issue: the relationship between persons with disabilities and social networks.
The study presents the findings from a cross-sectional survey on social and economic factors associated with the well-being of persons with disabilities in Kerala, India.
Although my knowledge about India is limited I believe that the methodology is appropriate.
Author response: Thank you for agreeing to review our manuscript and your suggestions have significantly improved the quality of the manuscript.
The study conducted a community-based survey across three geographical zones, North, Central, and South of Kerala state, randomly selecting two districts from each zone using a stratified sample method, followed by one local self-government from each of these six districts.
Also, the choice of allowing Community health professionals to identify individuals with disabilities, and mixing participants with physical an intellectual disabilities are optimal choices.
Thus, from a methodological perspective the article is sound.
Therefore, it surely represents a necessary contribution to the stream of literature looking at social network uses by disabled people.
Author response: Thank you
However, I believe that this work present one major flaws that the authors should fix: the article lacks a solid theoretical background
1) INTRODUCTION
The Introduction is somehow superficial and does not offer a solid background to justify the study.
I suggest dividing it in at least two subheading: the first dedicated –as you already do- to data about people with disabilities, maybe adding specificities of Kerala, to better justify the study.
The second should be dedicated to the wide and SEMINAL literature dealing with social networks and disability, that in the current state of the paper is not dealt with.
Author response: the second paragraph actually talks about the literature dealing with social networks of people with disability, we have added a couple more references to it. Hope it is clear now.
This should be the backbone of your paper that actually a) justify your work b) located in a stream of research
Author response: The introduction has been restructured to bring in changes as suggested.
In particular, I strongly encourage you to situate your study within the framework DIGITAL AND MEDIA LITERACY, since disability can be a form of social exclusion (as your own study demonster) that can be tackled by media literacy.
These studies might help:
Friesem, Y. (2017). Beyond accessibility: How media literacy education addresses issues of disabilities. Journal of Media Literacy Education, 9(2), 1-16.
Cervi, L.; Tornero, J.M.P. Changing the Policy Paradigm for the Promotion of Digital and Media Literacy. The European Challenge.In Pursuing Digital Literacy in Compulsory Education: Reconstructing the School to provide Digital Literacy for All; Peter Lang Inc.: NewYork, NY, USA, 2011; pp. 50–70.
Author response: It would be really interesting to have that perspective in our study. We will try to further carry on research from that perspective. Thank you for the suggestion. We have also discussed that possibility in the manuscript.
2) CONCLUSIONS
Once the theoretical background has been enriched in the Conclusion it would be important to engage with previous studies.
For instance, the rich social media and Covid-19 stream of literature might be used to both compare your results and underline their importance.
Some examples:
Tejedor, S.; Cervi, L.; Pérez-Escoda, A.; Tusa, F. Smartphone usage among students during COVID-19 pandemic in Spain, Italyand Ecuador. In Eighth International Conference on Technological Ecosystems for Enhancing Multiculturality, Salamanca, Spain, 21–23 October 2020; ACM: New York, NY, USA, 2020; pp. 571–576.
Dobransky, K., & Hargittai, E. (2021). Piercing the pandemic social bubble: Disability and social media use about COVID-19. American Behavioral Scientist, 65(12), 1698-1720.
Author response: We have included the message in the conclusion. Thank you for your valuable suggestions.
GOOD LUCK!
Reviewer 3 Report
Thank you for the opportunity to review this paper. Congratulations on a well-presented piece of work. I have made some minor suggestions.
Please update references.

Author Response
Thank you for the opportunity to review this paper. Congratulations on a well-presented piece of work. I have made some minor suggestions.
Please update references.
Thank you for the suggestions. The changes have been made to the manuscript.
- Line 173 - 147 (32.7%) of PWDs had mild and above depression, 93 (20.7%) had mild and above anxiety, and 278 (61.8%) had mild and above
- Details on “other disabilities” have been added in the footnote.
- Percentages have been added to wherever relevant
- Line 242 – the sentence “There is sufficient evidence to prove that people living in supportive communities require fewer mental health services [29] due to better well-being” – talks about the evidence in the literature, not the study finding - a reference is also added.
- More references have been added to the discussion section
- Reference style is updated, also newer references are added.
Round 2
Reviewer 2 Report
The authors have not implemented any changes to the manuscrip.
My suggestions have been ignored.
AGAIN:
1) INTRODUCTION
The Introduction is somehow superficial and does not offer a solid background to justify the study.
I suggest dividing it in at least two subheading: the first dedicated –as you already do- to data about people with disabilities, maybe adding specificities of Kerala, to better justify the study.
The second should be dedicated to the wide and SEMINAL literature dealing with social networks and disability, that in the current state of the paper is not dealt with.
Author response: the second paragraph actually talks about the literature dealing with social networks of people with disability, we have added a couple more references to it. Hope it is clear now.
This should be the backbone of your paper that actually a) justify your work b) located in a stream of research
Author response: The introduction has been restructured to bring in changes as suggested.
In particular, I strongly encourage you to situate your study within the framework DIGITAL AND MEDIA LITERACY, since disability can be a form of social exclusion (as your own study demonster) that can be tackled by media literacy.
These studies might help:
Friesem, Y. (2017). Beyond accessibility: How media literacy education addresses issues of disabilities. Journal of Media Literacy Education, 9(2), 1-16.
Cervi, L.; Tornero, J.M.P. Changing the Policy Paradigm for the Promotion of Digital and Media Literacy. The European Challenge.In Pursuing Digital Literacy in Compulsory Education: Reconstructing the School to provide Digital Literacy for All; Peter Lang Inc.: NewYork, NY, USA, 2011; pp. 50–70.
Author response: It would be really interesting to have that perspective in our study. We will try to further carry on research from that perspective. Thank you for the suggestion. We have also discussed that possibility in the manuscript.
2) CONCLUSIONS
Once the theoretical background has been enriched in the Conclusion it would be important to engage with previous studies.
For instance, the rich social media and Covid-19 stream of literature might be used to both compare your results and underline their importance.
Some examples:
Tejedor, S.; Cervi, L.; Pérez-Escoda, A.; Tusa, F. Smartphone usage among students during COVID-19 pandemic in Spain, Italyand Ecuador. In Eighth International Conference on Technological Ecosystems for Enhancing Multiculturality, Salamanca, Spain, 21–23 October 2020; ACM: New York, NY, USA, 2020; pp. 571–576.
Dobransky, K., & Hargittai, E. (2021). Piercing the pandemic social bubble: Disability and social media use about COVID-19. American Behavioral Scientist, 65(12), 1698-1720.
Author Response
The authors have not implemented any changes to the manuscript.
My suggestions have been ignored.
Response: We sincerely regret that you felt that way; we did try to incorporate your suggestions into our manuscript. Unfortunately, because we could not thoroughly explore this domain in our current work, we were unable to include this component adequately Additionally, we are also unsure of how this perspective will be applicable in Kerala’s socio-cultural scenario and with the early / older adults that made up the majority of respondents. Despite Kerala's high smartphone penetration rate (65%), digital literacy was found to be feasible with service delivery alone. The emotional component of social support and inclusion in Kerala continues to lie in one-on-one, in-person neighbourhood interactions. Nonetheless, it is a grey area worth exploring to leverage the digital and media literacy perspective. Thank you for this great suggestion, which would guide our future research.
We have made an effort to incorporate the suggestions in our revised manuscript.
INTRODUCTION
The Introduction is somehow superficial and does not offer a solid background to justify the study.
I suggest dividing it in at least two subheadings: the first dedicated –as you already do- to data about people with disabilities, maybe adding specificities of Kerala, to better justify the study.
The second should be dedicated to the wide and SEMINAL literature dealing with social networks and disability, that in the current state of the paper is not dealt with.
Author response: The introduction section is divided into two as suggested, and a few references have been added. The first paragraph has been modified to include specificities from Kerala as well.
This should be the backbone of your paper that actually a) justify your work b) is located in a stream of research.
Author response: The introduction has been restructured to bring in changes as suggested. Thank you for your suggestions. I think the suggested changes could improve the clarity and quality of the manuscript.
In particular, I strongly encourage you to situate your study within the framework DIGITAL AND MEDIA LITERACY, since a disability can be a form of social exclusion (as your own study demonstrate) that can be tackled by media literacy.
These studies might help:
Friesem, Y. (2017). Beyond accessibility: How media literacy education addresses issues of disabilities. Journal of Media Literacy Education, 9(2), 1-16.
Cervi, L.; Tornero, J.M.P. Changing the Policy Paradigm for the Promotion of Digital and Media Literacy. The European Challenge. In Pursuing Digital Literacy in Compulsory Education: Reconstructing the School to provide Digital Literacy for All; Peter Lang Inc.: NewYork, NY, USA, 2011; pp. 50–70.
Author response: It would be exciting to have that perspective in our study. We will try to carry on research from that perspective further. Thank you for the suggestion. We have discussed the possibility of incorporating this perspective into the manuscript. We would also like to potentially distinguish the European and Indian contexts with respect to social exclusion and integration. Thank you for the suggested readings, which have the potential to explore this dimension in our future research.
Also, media literacy has been used as an effective strategy for children with disabilities. Since these children are digital natives belonging to Gen-Z and Gen Alpha (as study reference 1 suggested), digital and media literacy would be a potentially effective strategy in India also. However, the majority of our respondents (65%) belonged to early / or middle adulthood, and 12.9% were elderly, we need to explore further the feasibility of this model with the digital migrant groups as well. Therefore, this suggestion will guide our future research in this area. Thank you for this thoughtful suggestion.
2) CONCLUSIONS
Once the theoretical background has been enriched in the Conclusion it would be important to engage with previous studies.
For instance, the rich social media and Covid-19 stream of literature might be used to both compare your results and underline their importance.
Some examples:
- Tejedor, S.; Cervi, L.; Pérez-Escoda, A.; Tusa, F. Smartphone usage among students during COVID-19 pandemic in Spain, Italyand Ecuador. In Eighth International Conference on Technological Ecosystems for Enhancing Multiculturality, Salamanca, Spain, 21–23 October 2020; ACM: New York, NY, USA, 2020; pp. 571–576.
- Dobransky, K., & Hargittai, E. (2021). Piercing the pandemic social bubble: Disability and social media use about COVID-19. American Behavioral Scientist, 65(12), 1698-1720.
Author response: Based on your suggestions, we have added a few points in the discussion section regarding how technology can improve social cohesion and potentially help PWDs.
We hope we have made the changes suggested by the reviewer and given adequate explanations for the same. I am extremely grateful to the reviewer who has been patient and helped us guide the research in a new direction, and I feel it helped improve the quality of the manuscript.